# FINE-GRAINED ENERGY PREDICTION FOR PARALLELIZED LLM INFERENCE WITH PIE-P

## ABSTRACT

With the widespread adoption of Large Language Models (LLMs), energy costs of running LLMs is quickly becoming a critical concern. However, precisely measuring the energy consumption of LLMs is often infeasible because hardware-based power monitors are not always accessible and software-based energy measurement tools are not accurate. While various prediction techniques have been developed to estimate LLM energy consumption, these approaches are limited to single-GPU environments and thus are not applicable to modern LLM inference which is typically parallelized across multiple GPUs. In this work, we remedy this gap and introduce PIE-P, a fine-grained energy prediction framework for multi-GPU inference, including tensor, pipeline, and data parallelism. Predicting the energy under parallelized inference is complicated by the non-determinism in inter-GPU communication, additional communication overheads, and difficulties in isolating energy during the communication/synchronization phase. We develop a scalable prediction framework that addresses these issues via precise sampling, fine-grained modeling of inter-GPU communication, and careful accounting of parallelization overheads. Our evaluation results show that PIE-P yields accurate and fine-grained energy predictions across parallelism strategies, significantly outperforming baselines.

## 1 INTRODUCTION

Scaling model capacity (Chen et al., 2025), i.e., increasing the number of model parameters, has vastly improved instruction following (Zhang et al., 2023), tool use (Qin et al., 2023), and reasoning abilities (Xu et al., 2025) in many Large Language Models [1] (LLMs) including Llama, Olmo, GPT4, Claude, and many others. However, these scaling improvements have come at the expense of corresponding *increase in compute and energy costs*. Coupled with the widespread adoption and deployments for web scale queries[2], these advances also raise ***LLM inference energy*** to staggering levels. Recent estimates show that energy consumption for LLM inference can account for 80–90% of total energy consumption in certain data centers (Wenjen, 2024), and the inference energy alone is projected to grow to 1,050 terawatt-hours by 2026 (Kakolyris et al., 2024).

Given the environmental and cost considerations of LLM energy use (Strubell et al., 2019; Schwartz et al., 2020; Bender et al., 2021; Morrison et al., 2025), it is critical to understand the energy consumption of LLM inference to design energy-efficient models. Knowing the model-level energy helps developers of LLM-based services make energy-aware choices in terms of model sizes, model optimizations, and parallelization strategies. Further, knowing the energy consumed by individual modules of an LLM can help identify key energy bottlenecks.

A natural approach to characterize LLM energy consumption is to measure it directly using full-system hardware energy monitors (Anthony et al., 2020; Courty et al., 2024a; Shaikh et al., 2021; Argerich & Patiño-Martínez, 2024). However, in multi-tenant clusters hardware power monitors are often unavailable, since they require inline meters, administrative privileges, and typically exclusive machine access. Finally, direct measurement techniques cannot measure the energy consumption of *individual components* of an LLM, such as at the module level.

---

[1] In this work, we consider LLMs to be models with billions of parameters.
[2] OpenAI services have hit a billion queries per day as of April 2025 (Singh, 2025).

As a result, the first critical step of accurately estimating the energy consumption of an LLM and understanding its energy bottlenecks remains challenging. This is especially true in ***parallelized-inference*** scenarios, a setting widely used today to overcome GPU memory (and compute) constraints. There has been related work on designing prediction frameworks to predict LLM energy consumption. These frameworks either use coarse-grained, resource utilization-based techniques (Strubell et al., 2019; Nguyen et al., 2024) or primarily consider GPU power consumption (Morrison et al., 2025). As a result, they do not capture the complex interactions that affect overall energy consumption and result in poor accuracy, as we demonstrate in our experiments. The most relevant approach, IrEne (Cao et al., 2021), does provide accurate energy predictions by employing a *model tree abstraction* that takes into account resource utilization and fine-grained model execution features for prediction. However, IrEne only works for a single GPU setting.

Parallel inference introduces significant new challenges for accurate energy prediction. First, energy consumed under parallel inference depends on the ***communication costs*** in addition to the computation and memory costs. Second, there is higher ***variance*** in energy consumption due to the inherent non-determinism in communication between the GPUs (Xiong et al., 2024). Finally, these challenges add to the basic generalization problem in energy prediction, where architectural variations between models prevent direct transferability of energy measurements and predictions, even within the same model family (Warraich et al., 2023).

In this work, we present **Parallelized Inference Energy Predictor (PIE-P)**, a fine-grained energy prediction framework designed for parallel LLM inference. PIE-P builds on the model-tree abstraction approach of IrEne (Cao et al., 2021) and breaks down the model into its constituent modules, and then predicts the energy consumption of the model as a function of the energy consumption of the modules. Unlike IrEne which is limited to single-GPU settings, however, PIE-P specifically applies to all three popularly deployed parallelism strategies—data parallelism (Hillis & Steele, 1986), pipeline parallelism (Choi et al., 2023), and tensor parallelism (Shoeybi et al., 2020). Tensor parallelism is the most challenging of the three settings; accordingly, our design focuses on this setting, but we then discuss the generalization of PIE-P to pipeline and data parallelism.

PIE-P addresses the challenges of energy prediction for multi-GPU parallel inference using the following key ideas: (i) *Synchronization Sampling* to mitigate non-determinism related measurement issues prevalent in tensor parallelism via a carefully designed sampling approach that records the energy measurements of idle times and uses it to accurately estimate energy spent during synchronization/GPU communication; (ii) *Use of Structural Model Features*, such as number of attention heads, to capture the relationship between model architecture and communication patterns when using different parallelisms; (iii) *Aggregate Runtime Feature Representation* to scalably and concisely represent the features of multiple GPUs based on aggregates of their runtime features; and (iv) *Expanded Model Tree Abstraction* to include the operations, such as AllReduce, across multiple GPUs and to capture inter-GPU communication overheads specific to different parallelisms. The expanded model tree abstraction is used to predict the energy consumption of the model and each LLM module. We emphasize that *both* module- and model-level prediction are essential: module-level profiling localizes optimization hotspots (e.g., attention/MLP and synchronization) while model-level prediction enables end-to-end energy estimation.

We implement PIE-P to predict the energy consumption on a variety of open LLM families (Vicuna (Chiang et al., 2023), Mistral (Jiang et al., 2023), Llama (Touvron et al., 2023), and Qwen (Bai et al., 2023)) of varying sizes (7B–70B). We use PIE-P's measurement methodology to obtain fine-grained energy measurements and build the prediction framework for these LLMs and their generalization across size, inputs, and model variants at both model and module levels for all three parallelisms. **We will release all code and data upon publication**. Our results demonstrate that PIE-P consistently achieves low model-level energy prediction error under tensor parallelism (MAPE $\approx 17.6\%$), pipeline parallelism (MAPE $\approx 13.25\%$), and data parallelism (MAPE $\approx 14.36\%$). Compared to the next-best performing baseline, PIE-P reduces energy prediction error by $1.5$–$3\times$. We end with a use case that shows how PIE-P can help navigate the trade-off between inference time and energy consumption per token across different model sizes and GPU configurations.

## 2 RELATED WORK

In recent years, there has been increased interest in the sustainability and energy consumption of LLMs, as well as improving the efficiency of LLMs. We describe these methods below and identify the gaps in the literature.

**LLM Energy modeling and prediction.** The seminal work by Strubell et al. (Strubell et al., 2019) and several follow up works (Nguyen et al., 2024; Wilkins et al., 2024) use coarse-grained resource-utilization or input/output tokens as a proxy to predict LLM energy consumption. The problem is that using utilization alone or tokens alone leads to poor prediction accuracy since it cannot capture the complex interactions that affect overall energy consumption (Cao et al., 2020); we demonstrate this experimentally in Section 5. The more recent work on energy prediction (Morrison et al., 2025) uses a tool called CodeCarbon (Courty et al., 2024b) which provides a lower bound prediction by accurately measuring power consumption of GPUs. Our experiments (Section 5) show that although this is more accurate than other approaches, it still results in large prediction errors.

The most related work to ours is IrEne (Cao et al., 2021), which takes into account the model structure and fine-grained model execution features to make accurate energy predictions for a model as well as its components. However, IrEne is not designed for parallel inference settings, and therefore results in increased prediction errors under these settings (see Section 5).

**Energy measurement and optimizations.** With the rising awareness of the environmental costs associated with LLMs, there has been a number of scholarly works on measuring energy consumption and estimating carbon emissions (Lacoste et al., 2019; Anthony et al., 2020; Courty et al., 2024a; Shaikh et al., 2021; Argerich & Patiño-Martínez, 2024; Luccioni et al., 2022). However, these works either use direct measurement with hardware energy monitors or use software-based energy profilers such as NVIDIA's NVML. As discussed earlier, hardware monitors are not always available for energy measurement. Further, for these measurements to be accurate, the model needs exclusive access to the hardware which is not always possible.

Software profilers such as NVML have poor accuracy as they only consider GPU power consumption and are widely treated as a *lower bound* on energy consumption; see NVIDIA; Lacoste et al. (2019); Anthony et al. (2020); Courty et al. (2024a); Shaikh et al. (2021); Argerich & Patiño-Martínez (2024); Luccioni et al. (2022). Consistent with this literature, in Appendices G and H we empirically confirm that software energy profilers underestimate energy, leading to large prediction errors.

Several existing works have also focused on LLM energy optimizations that include parallelization strategies, dynamic schedulers, and DVFS (dynamic voltage frequency scaling) (Kakolyris et al., 2024; Wilkins et al., 2024; Jia et al., 2024). Similar to energy optimization, there are also related works on optimizing LLM inference latency (Agrawal et al., 2025; Cheng et al., 2024; Dong et al., 2024). But these works do not extend to predicting energy and are addressing an orthogonal problem.

**Modeling and predicting latency.** More recent works (Lu et al., 2023; Hu et al., 2022; Luo et al., 2022; Lee et al., 2025; Yu et al., 2021; Zhu et al., 2020) have focused on predicting the latency of LLMs by tracing ML operations. However, predicting energy consumption is a fundamentally different problem compared to latency. For example, GPUs are optimized specifically for performance; as a result, scaling different architectural knobs (more cores, higher clocks) translates to lower latency in a relatively deterministic manner. Energy, on the other hand, does *not* scale linearly with the same knobs, making prediction much harder.

## 3 BACKGROUND AND CHALLENGES

Our goal is to develop a methodology for predicting the energy consumption of LLM inference over multiple GPUs. There are three main parallelism strategies that are widely used for LLM inference, described below in increasing order of complexity.

1. ***Data parallelism*** replicates the entire LLM on multiple GPUs, allowing input data to be processed in parallel (Hillis & Steele, 1986). However, this requires the LLM to be small enough to fit in the memory of a single GPU. At inference time, the outputs (e.g., logits or token probabilities) are combined using a strategy called *AllGather*. In an AllGather across $p$ replicas, each GPU

exchanges partitions of its tensor with others over $p-1$ steps (e.g., a ring implementation) so that eventually every GPU holds the concatenation of all replicas' outputs; see Appendix E for details.

2. **Pipeline parallelism** partitions the LLM into $s$ sequential stages, assigning each stage to a different GPU and processing micro-batches in a pipeline (Choi et al., 2023). This allows inference with models that do not fit in a single GPU's memory. During inference, communication is *point-to-point*: the activations produced by stage $i$ are sent to stage $i+1$ (and so on) for the next computation step; see Appendix D for the brief cost model and schematic.

3. **Tensor parallelism** splits individual model computations (such as matrix multiplications) across multiple GPUs to enable parallel execution of operations even within a single layer (Shoeybi et al., 2020). Similar to pipeline parallelism, this is used when the LLM is too large to fit on a single GPU, and is widely used because of its efficiency. One of the main additions to the LLM inference due to tensor parallelism is the *AllReduce* (Zong et al., 2025) component. AllReduce enables GPUs to sequentially communicate and reduce partially computed results across GPUs using *ring* protocol; the aggregated results are redistributed for further processing. These collation operations require inter-GPU communication and careful synchronization to ensure correct and consistent aggregation of partial results (see Appendix B).

**Challenges.** LLM energy consumption under parallel inference is driven by not only the computation cost, but also the inter-GPU communication costs. Under data parallelism, the replicas work independently, except for the collation of output from each replica in the AllGather phase; as such, the additional energy of this phase needs to be accounted for. The AllGather phase is largely deterministic, so predicting the energy consumption under data parallel inference is feasible. For pipeline parallelism, we have to track the energy consumption when data is sent from one stage to the other, in addition to the local computation. These data transfers happen in sync with each stage's computations and the timing can vary depending on sequence length and network speeds.

Tensor parallelism is the most challenging of the three cases because synchronization across GPUs (e.g., ReduceScatter/AllGather) is interleaved with layer computation for high efficiency. This leads to two main challenges: (i) *Non-determinism* in GPU wait times as GPUs often lag/lead each other in compute operations (due to variations in memory access, caching effects, hardware scheduling) during the synchronization phase. From a measurement standpoint, this causes difficulty in isolating AllReduce energy as it requires distinguishing between the non-deterministic waiting phase (where GPUs are idle) and the network transfer phase. (ii) *Fine-grained energy accounting* is challenging as AllReduce overlaps computation and network transfer. From a prediction standpoint, this requires isolating the different sources of energy consumption and accounting for them. This is not an issue in pipeline and data parallelism as GPU communication and computation are not synchronized.

## 4   DESIGN OF PIE-P

We now describe how we address the challenges above to design an accurate energy prediction framework. As a first step, we develop a fine-grained measurement methodology to measure communication energy (in addition to the computation energy) that will be used to train the prediction framework. We note that PIE-P performs all these measurements *offline*. We run repeated, controlled passes to capture the distributions of time and energy induced by GPU communication; these empirical distributions are then reused during prediction. As a result, during inference, PIE-P incurs no additional overhead.

Our prediction methodology builds on IrEne (Cao et al., 2021) and develops a multi-level regression model that uses fine-grained model description and resource utilization features. The key idea in IrEne is that the energy of a model can be predicted as a composition of the (predicted) energies of its components. To this end, IrEne builds a *model tree* abstraction that captures the model's computational structure; see details in Appendix A. However IrEne is designed for single GPU settings and does not work well for parallel inference (see Section 5).

PIE-P makes three significant extensions to IrEne's prediction methodology to handle parallel inference: (i) **expands the model tree abstraction** with dedicated communication modules—*AllReduce* for tensor parallelism, *point-to-point* stage transfers for pipeline parallelism, and *AllGather* for data parallelism—to capture inter-GPU synchronization/transfer overheads; (ii) **employs scalable, aggregate features** that capture the energy impact of multiple GPUs and replica/stage interactions; and (iii) uses an **expanded set of model description features** to capture the relationship between the

model architecture and communication energy under each parallelism strategy. Figure 1 shows the overall architecture of PIE-P. We explain the key components of PIE-P in the subsections below.

**Fine-grained Measurement.** Different from IrEne, we isolate the energy for GPU synchronization via precise energy measurement of computation and communication stages.

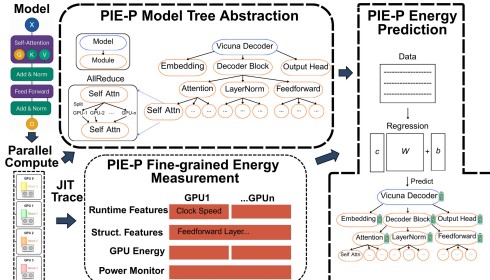

- Tensor Parallelism: During profiling, we capture the *full distribution* of energy consumption caused by non-deterministic GPU synchronization through multiple runs. By capturing a distribution of GPU wait times, rather than relying on a single execution run, we ensure that PIE-P reflects both leading and lagging GPU behavior, accounting for any variability.
- Pipeline Parallelism: We capture *point-to-point* transfer by benchmarking the interval between completion of boundary layer in the producing stage and start of the next stage in the next GPU.
- Data Parallelism: Since each replica produces its own output, profiling the final output stage already includes the terminal single AllGather.

Figure 1: PIE-P architecture consisting of three main components: (1) Energy Measurement captures energy consumption during computation and synchronization phases through specialized profiling, (2) Model Tree Abstraction that abstracts the model into a model tree, and (3) Energy Prediction that uses the model tree abstraction to learn energy consumption of the total model and each module.

These energy readings are then synchronized with system-level utilization logs (including GPU compute activity and memory bandwidth) to enable energy attribution to modules, such as AllReduce, within the model. For training, we collect energy measurements at the module level.

Runtime features including CPU metrics and GPU energy are collected using low-overhead libraries, NVIDIA-SMI and Linux's procfs. Structural features are extracted from the model architecture, and FLOPs are computed using standard formulas based on model dimensions and operations. We then use statistical aggregates such as min, max, standard deviation, and mean over these features, along with the number of GPUs provided as input, to make predictions.

**PIE-P Model Tree Abstraction.** The AllReduce module is integrated into the model tree abstraction at precise synchronization points within the tensor-parallelized transformer architecture. Specifically, we add nodes in the model tree abstraction after: (1) the self-attention *output* projection and (2) the MLP layer in the feed-forward network. Our profiling framework timestamps three critical phases: the initiation of the waiting phase (when fastest GPUs become idle), the beginning of network transfer (when actual data movement starts), and the synchronization completion (when all GPUs have finished processing).

**For pipeline parallelism**, the model tree abstraction also includes inter-stage transfer nodes at each pipeline boundary. We timestamp (1) completion of boundary layer in the producing stage, (2) the first byte placed on the interconnect, and (3) the start of first operation in the consuming stage. **For data parallelism**, the final output aggregation is represented as the batch-output module; profiling this module already captures the terminal AllGather, so additional synchronization nodes are not needed.

**PIE-P Features.** PIE-P specifically extends IrEne's feature set to accommodate the requirements of parallelized inference. Table 1 lists all features used in PIE-P, categorizing them into three groups: resource utilization, execution parameters, and model structure. PIE-P incorporates model structure features to capture the relationship between model architecture and communication patterns in tensor-parallel settings. For instance, the number of attention heads

**Resource Utilization Features**

CPU utilization (%)
CPU memory utilization (%)
GPU utilization (%)
GPU memory utilization (%)
CPU clock speed (GHz)
CPU memory clock speed (GHz)
Memory (bytes)
GPU clock speed (GHz)
GPU memory clock speed (GHz)

**Execution Features**

Batch size
Sequence length
FLOPs per token (billions)
Execution time (s)
GPU energy from NVML (Wh)
Number of GPUs*

**Model Structure Features***

Feed-forward dimension
Transformer blocks
Hidden embedding size
Attention heads
Key–value heads

Table 1: Features used by PIE-P for prediction; * denotes new features added for PIE-P.

influences how work is divided across GPUs during tensor parallelization, affecting communication volume in AllReduce operations. Similarly, the feed-forward dimension and hidden embedding size determine the size of tensors that must be communicated during synchronization.

Next, PIE-P obtains features from multiple GPUs. However, maintaining separate feature sets *for each GPU* is not a scalable solution and would create inconsistent feature dimensions for different parallelization degrees. Instead, PIE-P computes and employs ***statistical aggregates*** of the runtime features by computing the mean, standard deviation, min and max values across all GPUs. This scalable aggregation approach captures workload imbalance through the standard deviation and min/max statistics.

Finally, PIE-P incorporates GPU energy reported by NVML as part of its runtime feature set. The **feature set used for prediction is the same across tensor parallelism, pipeline parallelism, and data parallelism**. The significance of these feature choices is validated via a correlation analysis of runtime features (Appendix K) and an ablation study of model features (Appendix N).

**PIE-P Prediction.** PIE-P uses a multi-level regressor. However, unlike IrEne which builds its model tree from low-level ML primitives (e.g., Linear, Softmax) as the leaf nodes, PIE-P constructs the model abstraction tree directly at the module level (e.g., Self-Attention, Feed-Forward). This is because, for example, Tensor parallelism operates at the module level, where operations such as Self-Attention and Feed-Forward Networks (MLP) are split across GPUs. The energy consumption patterns emerge from how these higher-level modules communicate across GPUs, rather than from individual ML primitives.

$$
\begin{aligned}
P_e(n) &= \sum_{c \in child(n)} \alpha(c)\, P_e(c), && \text{if } n \text{ is non-leaf} \\
&= P_e^{\text{Module}_i}(n), && \text{if } n \text{ is leaf} \\
\alpha(c) &= 1 + \tanh\big(\mathbf{W}\, \text{feat}(c) + b\big)/\tau
\end{aligned} \tag{1}
$$

The regressor is specified via the energy prediction $P_e(n)$ for an arbitrary node $n$ in the model tree abstraction. The energy of a node $n$ is computed as a weighted sum of the energy of its children $c \in child(n)$. The weight $\alpha(c)$ for each child is itself predicted as a regression over the features $feat(c)$ of the corresponding component or through a separately trained regressor if the child $c$ is a leaf. Formally, the predicted energy for node $n$ is given in Eq. 1, where $\mathbf{W}$ are the recurrent parameters (weights) of the model learnt over a training set of ground-truth energy measurements and $P_e^{\text{Module}_i}(n)$ denotes the predicted energy for leaf node $n$ corresponding to module type $\text{Module}_i$, obtained using a module-specific regression.

## 5 EVALUATION

Our evaluation demonstrates PIE-P's effectiveness across diverse scenarios and across model families to validate its key technical components. Our evaluation primarily focuses on the most challenging case of tensor parallelism; the difficulties in energy prediction for this case were discussed in Section 3. Results for pipeline parallelism and data parallelism are discussed in Section 5.3.

**Evaluation Methodology.** Our experiments are conducted on a server with an AMD EPYC Milan 7543P processor (32 cores) and four NVIDIA RTX A6000 GPUs (48GB GDDR6, PCIe 4.0). Ground-truth system power/energy is measured with an external power monitor (Watts Up Pro). We apply the PIE-P methodology to obtain fine-grained measurements for four widely-used, diverse set of transformer-based LLM families across varying sizes (7B–70B): Vicuna, Mistral, Llama, and Qwen (Chiang et al., 2023; Jiang et al., 2023; Touvron et al., 2023; Bai et al., 2023). Their open-source nature allows for complete instrumentation and detailed energy profiling across different configurations. Models exceeding single-GPU memory (Vicuna-33B, Mistral-48B, Qwen-32B, Llama-70B) were tested only on multi-GPU configurations, with Llama-70B requiring 4 GPUs. We use the energy measurements data to then train and evaluate the PIE-P energy prediction model using 3-fold cross-validation; details of our training methodology are provided in Appendix L.

**Baselines.** We evaluate PIE-P against three baselines techniques:
(i) **IrEne** (Cao et al., 2021). We extend IrEne to multi-GPU by using aggregated runtime features, similar to PIE-P. Note that IrEne does not include inter-GPU collectives from model-level energy prediction; this helps isolate the benefits of PIE-P in explicitly modeling multi-GPU communication.
(ii) **CodeCarbon** (Courty et al., 2024b). We use CodeCarbon's measurement path that estimates energy from readily available telemetry (e.g., GPU via NVML, CPU via RAPL/powermetrics/heuristics)

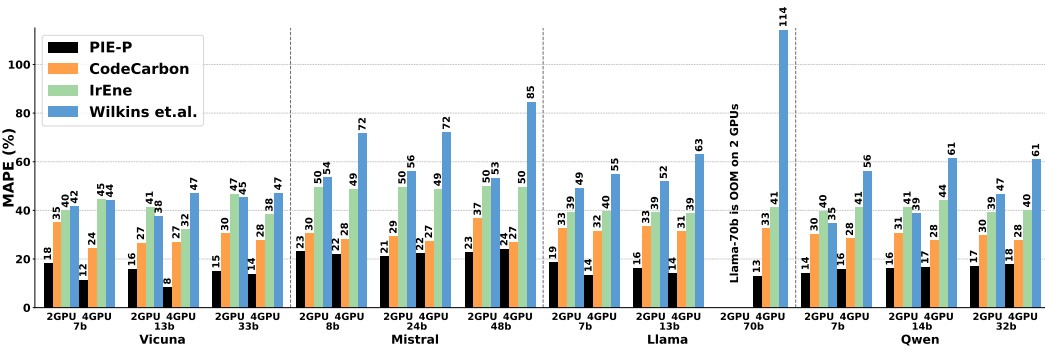

Figure 2: MAPE results across model families for PIE-P and the baselines under tensor parallelism.

and—optionally—maps energy to $CO_2$ via grid carbon intensity; this is chosen because it is widely used by researchers and practitioners for system-level energy/emissions estimation, including more recently by Morrison et al. (2025).

(iii) **Wilkins et al.** (Wilkins et al., 2024). We implement this token-in/token-out regression technique which predicts per-request energy as a function of input and output token counts with an interaction term in Equation 2, where the coefficients

$$e_K(\tau_{\mathrm{in}}, \tau_{\mathrm{out}}) = \alpha_0 \, \tau_{\mathrm{in}} + \alpha_1 \, \tau_{\mathrm{out}} \\ + \alpha_2 \, \tau_{\mathrm{in}} \tau_{\mathrm{out}} \quad (2)$$

$(\alpha_0, \alpha_1, \alpha_2)$ are fit from a calibration set. This is included as a recent, deployment-friendly approach that captures length-based energy variation without hardware monitors.

For completeness, we also evaluate PIE-P *without* the waiting phase (i.e., our model tree and structural features without synchronization sampling) as an ablation study in Appendix J to highlight the importance of synchronization sampling in PIE-P.

## 5.1 ENERGY PREDICTION RESULTS FOR TENSOR PARALLELISM

Figure 2 shows the mean absolute percentage error (MAPE) for PIE-P and the three baselines for all four model families across model sizes and parallelization degrees. For each model family (e.g., Vicuna), we train a regressor on 70% of module-level predictions aggregated across all variants (e.g., 7B) and evaluate on the remaining 30%. Each bar in Figure 2 reflects the resulting MAPE for one variant, with baseline comparisons included (see Appendix L for training details).

We see that *PIE-P consistently achieves the lowest MAPE in all cases*, with an average of 17.6%. We also observed that removing synchronization sampling from PIE-P substantially increased MAPE from 17.6% to 36.9%, suggesting the importance of fine-grained sampling in the design of PIE-P. See Appendix J for the ablation and discussion.

By contrast, CodeCarbon incurs an average MAPE of 28.49%, which is about $1.7\times$ the MAPE of PIE-P. The poor prediction accuracy of CodeCarbon is likely because of its reliance on coarse GPU profiling and slow sampling, which misses the fine-grained multi-GPU sync/transfer events and their energy consumption. IrEne also performs poorly, with an average MAPE of 40.45% (about $3\times$ that of PIE-P). This is to be expected as IrEne prediction is designed for single-GPU settings and omits inter-GPU communication, which leads to systematic misattribution under parallelism. Finally, Wilkins et al. consistently incurs high MAPE, with an average of 58.77% (more than $4\times$ that of PIE-P). This is because it solely uses a token-in/token-out regression that ignores inter-GPU communication. It also ignores hardware and runtime variance, and so its accuracy worsens with the degree of parallelism (number of GPUs employed).

On closer inspection of Figure 2, we also see that the *performance gap between PIE-P and the baselines widens with increasing parallelization*. This is because the energy contribution of AllReduce, due to its ring communication topology (see Section 3), increases with the number of GPUs; see Figure 5 in Appendix C. By ignoring this important AllReduce component, the baselines incur systematic prediction errors that worsen with parallelization. In contrast, across models, scaling from 2 to 4 GPUs causes only a modest change in MAPE for PIE-P, validating its robust predictions.

We also find that ***prediction accuracy depends on model complexity***. For instance, the PIE-P accuracy for Vicuna and Llama is superior (average MAPE of 14.3% and 15.4%, respectively) compared to that for Mistral and Qwen (average MAPE of 22.4% and 16.5%, respectively).

This disparity correlates with the corresponding transformer block complexities, as shown in Table 2. Mistral's higher MAPE (particularly at 22%–24% for the 48B variants) can be attributed to its more sophisticated grouped-query attention mechanism and SwiGLU activation, which generate more complex communication

| Model | MAPE | FLOPs/Block | Modules/Block |
|--------|--------|-------------|----------------|
| Vicuna | 6.28% | 187 GFlops | Standard Self-Attn., MLP |
| Mistral | 11.01% | 245 GFlops | Grouped-Query Attn., SwiGLU |
| Llama | 7.93% | 203 GFlops | Rotary Embeddings, RMSNorm |
| Qwen | 9.03% | 213 GFlops | Multi-Query Attn., Rotary |

Table 2: Energy prediction errors at the transformer module level across model families; models with more complex attention mechanisms (e.g., Mistral, Qwen) tend to show higher errors.

patterns during synchronization. Similarly, Qwen's multi-query attention mechanism contributes to its slightly elevated prediction error compared to Vicuna's relatively simpler architecture. As part of future work, we will develop models that capture these complex interactions for improved accuracy. For detailed module-level prediction results, please refer to Appendix F.

**Generalization across unseen variants.** To evaluate PIE-P's ability to predict the energy of unseen model variants—a critical capability for practical deployment—we conduct extensive Leave-One-Out cross-validation experiments across model and batch sizes. The methodology is further detailed in Appendix L. Table 3 shows model-level prediction errors for all model families.

We see that *PIE-P generalizes well to model sizes*. Specifically, PIE-P maintains reasonable accuracy across all families when trained on most model sizes and tested on a held-out size. The MAPE ranges from 15.84% (Vicuna 7B) to 24.52% (Mistral 48B), with an average generalization MAPE

| Model | MAPE | Model | MAPE |
|--------|--------|--------|--------|
| Vicuna 7B | 15.84% | Llama 7B | 18.15% |
| Vicuna 13B | 17.72% | Llama 13B | 21.11% |
| Vicuna 33B | 17.55% | Llama 70B | 22.41% |
| Vicuna BS-16 | 16.89% | Llama BS-16 | 18.33% |
| Vicuna BS-32 | 18.20% | Llama BS-32 | 19.16% |
| Mistral 8B | 21.17% | Qwen 8B | 20.15% |
| Mistral 24B | 23.13% | Qwen 14B | 20.99% |
| Mistral 48B | 24.52% | Qwen 32B | 18.98% |
| Mistral BS-16 | 19.79% | Qwen BS-16 | 20.15% |
| Mistral BS-32 | 21.82% | Qwen BS-32 | 19.15% |

Table 3: Leave-one-out prediction results for PIE-P. One model size or batch size (BS) is excluded from training and used only for testing.

of 19.99% across all model sizes. *PIE-P also generalizes well over batch sizes* with MAPE values ranging from 16.89% to 21.82%, with an average of 19.05% across all families. For detailed results, refer to Appendix I.

The consistent performance across these leave-one-out experiments shows that PIE-P learns generalizable relationships between model architecture and energy patterns in tensor parallel environments. This is especially useful for predicting energy consumption of new model variants or batch sizes without expensive profiling campaigns.

**Generalization across models.** We also evaluate PIE-P's ability to generalize across entire model architectures—a significantly more challenging task. We exclude one model family from training and evaluate predictions on that held-out family. Table 4 presents these cross-model results.

When generalizing to the Vicuna family (first row in Table 4), PIE-P achieves a MAPE of 24.1%, compared to 49.3% for IrEne, a relative improvement of 51.1%. This pattern holds for all families. In practical deployment scenarios where new model architectures are introduced, PIE-P's cross-architecture generalization capability, demonstrated in Table 4, represents a significant advantage, allowing reasonable energy predictions for novel model families without requiring extensive retraining or measurement.

| Excluded family | PIE-P | IrEne |
|------------------|--------|--------|
| Vicuna | 24.1% | 49.3% |
| Mistral | 27.0% | 56.5% |
| Llama | 26.1% | 55.3% |
| Qwen | 27.6% | 58.4% |

Table 4: Cross-architecture generalization results. Each row lists the model excluded from training.

## 5.2 USE CASE: INFERENCE TIME VS ENERGY FOR DIFFERENT LEVELS OF PARALLELIZATION

Consider a setting where an LLM user needs to choose a model and the number of GPUs across which to deploy. In addition to maximizing throughput (i.e., have less time spent on inference per

token), lower energy consumption is also an important consideration. We show how the user can employ PIE-P to make the right decision. Figure 3 illustrates the trade-off between (measured) inference time per token and predicted energy consumption per token across different Vicuna model sizes and GPU configurations under tensor parallelism; we use the highest batch size achievable for each model size. We verify that this trend remains consistent when using actual energy values (see Figure 8 in Appendix M).

While energy increases at maximum throughput, both per-token energy cost and inference time decrease with the degree of parallelization; this is true for all three model sizes. However, as the model size scales from 7B to 33B, energy rises due to increased computational complexity, leading to diminishing efficiency gains from paralleliza-tion. In effect, the energy impact of parallelization is not straightforward, and having predicted energy can help the user make an informed choice.

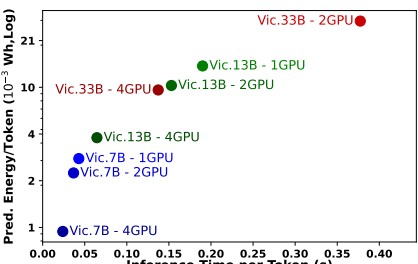

Figure 3: Predicted trade-off between inference time per token and energy per token (on log scale) for Vicuna under tensor parallelism.

### 5.3 ENERGY PREDICTION RESULTS FOR PIPELINE AND DATA PARALLELISM

Figure 4 shows the MAPE for PIE-P and the base-lines for the Vicuna family across model sizes and parallelization degrees under pipeline and data par-allelism. We omit the Wilkins et al. baseline as it is significantly inferior to others (see Figure 2). We do not report Vicuna 33B results for data parallelism as this model size does not fit in the GPU memory.

Similar to the tensor parallelism results, ***PIE-P con-sistently achieves the lowest MAPE in all cases***, with an average of 14.84% and 15% for pipeline and data parallelism, respectively. CodeCarbon has more than 2× the prediction error of PIE-P, with an average

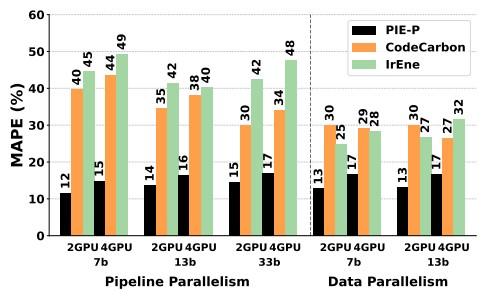

Figure 4: MAPE results for Vicuna under pipeline and data parallelism.

MAPE of 36.8% and 30.25% for pipeline and data parallelism, respectively. IrEne incurs average MAPE of 45.6% and 28% for pipeline and data parallelism, respectively. This is because both CodeCarbon and IrEne ignore communication energy during inference; however, this hurts less under data parallelism because here the communication occurs in a single AllGather stage. In contrast, under pipeline parallelism, data is transferred point-to-point repeatedly, resulting in larger errors when omitting communication energy.

## 6 CONCLUSIONS AND LIMITATIONS

Understanding the energy costs of LLMs is a critical first step towards energy efficient design and deployment but one that is also challenging. We present PIE-P, a prediction framework that remedies a key gap in accurate energy estimation of LLMs in parallelized inference settings, a necessity as model sizes demand multi-GPU inference. PIE-P provides a new measurement method that overcomes challenges that stem from non-determinism and synchronization related aspects of parallelizing inference. For accurate prediction, PIE-P expands tree abstractions of the model and adds structural model descriptors to a multi-level regressor. Empirical results across models show that PIE-P significantly reduces prediction error compared to previous approaches for all three parallelization strategies.

We acknowledge that PIE-P is hardware-dependent, and one of our immediate next steps is to bridge this hardware dependency gap. Through our experiments, we have found hardware-agnostic energy prediction for LLMs to be a challenging task, warranting a full project in itself. PIE-P currently supports only decoder-style transformer models. Extending it to encoder-based or encoder-decoder models is left for future work. This is challenging because encoder and encoder-decoder models often have bidirectional attention patterns and different execution flows, which affect both the structure of the model tree and the communication patterns.

# 7 REPRODUCIBILITY STATEMENT

We have taken steps to support reproducibility. The paper specifies the prediction method and design choices in section 3 and section 4, respectively, the experimental setup and evaluation protocol in section 5, and ablations and edge cases (Appendix J and additional appendices). Upon acceptance, we will release the code of PIE-P as supplementary material, including scripts for (i) offline profiling, (ii) feature extraction, (iii) model training/evaluation, and (iv) figure/table generation. We provide the details of our experimental setup in section 5. The configuration files with instructions to reproduce our results will be released with the code along with datasets/workloads, prompts, and execution modalities used in each experiment.

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

# APPENDIX

## A  BACKGROUND: IRENE BASELINE

IrEne (Cao et al., 2021) is a multi-level energy prediction model that combines fine-grained model execution features with resource utilization for more accurate energy prediction. IrEne first constructs a model tree abstraction that captures the model's computational structure at three levels: math level, machine learning (ML) level, and module level. The math level consists of basic mathematical operations like matrix multiplication and addition, which are model-agnostic and can generalize across various models. The operations at the ML level nodes, such as Linear and LayerNorm, are more specific to ML tasks but are still generalizable. The module level nodes (e.g., SelfAttention) are higher-level nodes that are made up of several ML tasks combined together.

IrEne then uses a bottom-up prediction approach for energy estimation. At the leaf level, IrEne learns the energy consumption of each node using features derived from runtime resource utilization (e.g., GPU usage) and model specifications (e.g., input size). IrEne uses ground truth measurement for training. At each intermediate node going up to the root, energy predictions are recursively computed by combining the predicted energy values of child nodes through a single regressor.

## B  BACKGROUND: ALLREDUCE

PIE-P extends the model tree abstraction of IrEne by introducing a dedicated module for AllReduce operations, which is essential for capturing the energy dynamics of tensor-parallelized inference. This extension is necessary because AllReduce operations introduce unique energy consumption patterns tied to inter-GPU communication that cannot be captured by the existing math, ML, or module level nodes.

The AllReduce module in our model tree abstraction represents the ring-based communication pattern that is fundamental to tensor parallelism. In the ring AllReduce algorithm, GPUs are arranged in a logical ring structure where each GPU communicates only with its immediate neighbors, reducing global communication overhead. The operation proceeds in two distinct phases:

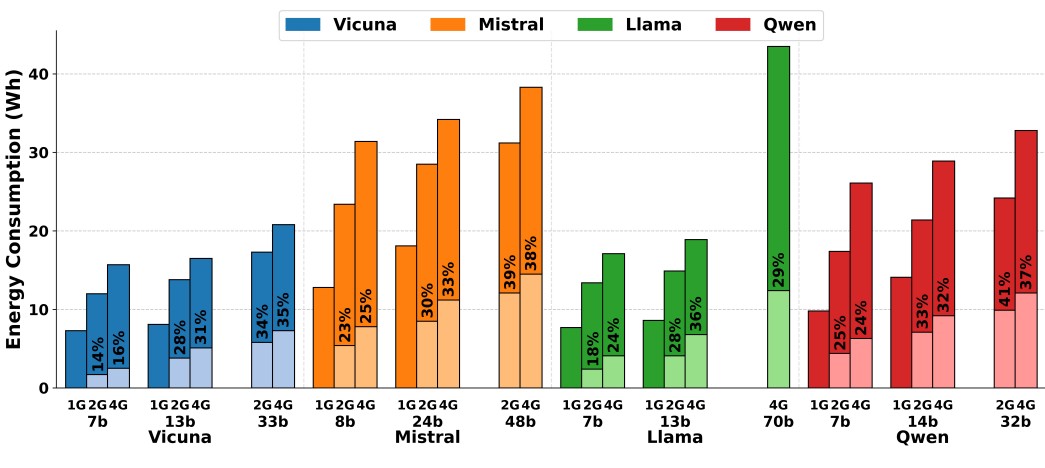

Figure 5: Energy consumption across different model families and GPU configurations. The bars show total energy consumption per model with lighter-colored segments representing AllReduce communication energy (% shown) and darker segments showing remaining computation energy.

1. **ReduceScatter Phase**: Each GPU divides its data into equal chunks (one per GPU). In this phase, each GPU sends a chunk to its successor in the ring and receives a chunk from its predecessor. Upon receiving data, each GPU performs a reduction operation (e.g., sum) with its corresponding local chunk. This process continues for $n - 1$ steps (where $n$ is the number of GPUs), at which point each GPU has one fully reduced chunk of the final result.

2. **AllGather Phase**: In this phase, the reduced chunks are disseminated to all GPUs. Each GPU sends its reduced chunk to its successor and receives a different reduced chunk from its predecessor. After $n - 1$ steps, all GPUs have all chunks of the fully reduced data.

There are two phases of the AllReduce process that is captured by the model abstraction tree:

1. **Communication Nodes**: Capturing the data transfers between adjacent GPUs in the ring.

2. **Synchronization Nodes**: Modeling the waiting periods where faster GPUs idle while waiting for slower ones to complete their operations.

## C  ADDITIONAL RESULTS: ALLREDUCE ENERGY CONSUMPTION

Figure 5 shows the energy consumption breakdown for parallelized inference across different GPU configurations for four model families: Vicuna, Mistral, Llama, and Qwen. Across all model families, AllReduce consumes a considerable amount of energy both for 2 to 4 GPUs in tensor-parallel configurations for batched inferences. For example, for the Vicuna family, the AllReduce energy constitutes 14.2% (1.7 Wh/12.0 Wh) of total energy with 2 GPUs for the 7B model, increasing to 15.9% (2.5 Wh/15.7 Wh) with 4 GPUs. This proportion is higher for larger models, reaching 27.5% (5.8 Wh/17.3 Wh) for Vicuna-33B with 2 GPUs and 35.1% (7.3 Wh/20.8 Wh) with 4 GPUs.

We observe that the proportion of AllReduce energy correlates strongly with the complexity of the transformer architecture. Models with more sophisticated attention mechanisms and larger feed-forward dimensions, such as Mistral and Qwen, show significantly higher AllReduce energy proportions compared to models with simpler architectures like Vicuna. More complex transformer blocks lead to higher AllReduce energy proportions because they generate larger intermediate tensors that must be synchronized across GPUs. For instance, Mistral-8B employs grouped-query attention with 32 attention heads but only 8 key-value heads, resulting in approximately 245 GFLOPs per transformer block—31% higher than Vicuna-7B's 187 GFLOPs per block with standard self-attention and 32 attention heads. As a result, the proportional energy consumption of AllReduce for Mistral is higher.

## D    POINT-TO-POINT DATA TRANSFERS IN PIPELINE PARALLELISM

In pipeline parallelism, stage $i$ sends its forward activations to stage $i+1$ via *point-to-point* device-to-device transfers (e.g., NVLink/PCIe/Ethernet). During inference there are no gradients; the only communication is these hop-local activation sends/receives. Operationally, the transfer for a given microbatch occurs immediately after stage $i$ finishes its forward compute and before stage $i+1$ launches its first kernel on the received activations. We profile this communication by timestamping (with nsys) the end of stage $i$'s last forward kernel and the start of stage $i+1$'s first forward kernel; the interval between these two events is attributed to the Point-to-Point transfer. We integrate power over this window to obtain P2P energy, repeating across hops and microbatches to form robust aggregates. Because these are explicit, hop-local sends/recvs rather than collectives, timing variability is typically small, and the attribution is more deterministic than in tensor-parallel synchronization phases.

## E    ALLGATHER COLLATION IN DATA PARALLEL INFERENCE

In data parallelism, each replica computes the full forward pass for its local microbatch and then *collates the final outputs* (e.g., logits/token scores) across replicas with a single AllGather at the tail of inference. Mechanistically, the last layer on each GPU (typically the output projection / logits head) finishes, the runtime immediately launches an NCCL AllGather to exchange each replica's output tensor, and the concatenated result is handed to the host for post-processing (sampling, metrics, or write-out). This makes profiling straightforward: *profiling the output layer automatically profiles the AllGather*, since it is the next and final step. This collation is a *single, tail-end* exchange of *final outputs* (logits/token scores) whose tensors are much smaller than hidden activations, and it happens once per request (or per decode step).

## F    MODULE-LEVEL ENERGY PREDICTIONS

Table 5 shows the module-level energy prediction MAPE achieved by PIE-P for the primary modules of Vicuna. For both 2-GPU and 4-GPU configurations, PIE-P achieves low errors for the compute-intensive Self-Attention (8.8–11.4%) and MLP modules (6.6–9.5%). For the communication-intensive AllReduce, the MAPE afforded by PIE-P is relatively higher (17.3–19.4%) owing to the greater variance in inter-GPU communication overheads and non-deterministic delays; nonetheless, the prediction errors remain reasonable for reliable modeling (see Appendix L for implementation).

| Module | 2 GPUs | 4 GPUs |
|---|---|---|
| Self-Attention | 8.8% | 11.4% |
| MLP | 6.6% | 9.5% |
| AllReduce | 17.3% | 19.4% |
| LayerNorm/RMSNorm | 6.4% | 7.3% |
| LLMEmbedding | 9.9% | 9.6% |

Table 5: **Module-level MAPE.** PIE-P energy prediction across components in 2-GPU and 4-GPU settings.

## G    NVML AS A PROXY FOR TOTAL ENERGY

Hardware vendors provide built-in power measurement interfaces such as NVIDIA Management Library (NVML) that report GPU-only energy consumption. Can these readily-available measurements be reliable proxies for total system energy through simple regression techniques?

Table 6 shows the results when NVML is used to predict total energy. Prediction errors across all model families and sizes are high. Vicuna models' MAPE range from 29.8% to 33.4%, while Mistral's are even higher between 39.0% and 44.2%.

| Model | MAPE | Model | MAPE |
|---|---|---|---|
| Vicuna 7B | 33.4% | Llama 7B | 31.1% |
| Vicuna 13B | 31.4% | Llama 13B | 31.3% |
| Vicuna 33B | 29.8% | Llama 70B | 28.5% |
| Mistral 8B | 44.2% | Qwen 8B | 42.4% |
| Mistral 24B | 40.3% | Qwen 14B | 38.0% |
| Mistral 48B | 39.0% | Qwen 32B | 38.3% |

Table 6: MAPE when using NVML-reported GPU energy as a proxy for total system energy. This approach produces substantially higher errors compared to PIE-P.

NVML also generalizes poorly to unseen models, as shown in Appendix H. The result shows that tensor-parallelized inference energy involves complex interactions between computation, synchronization, and system-level dynamics that cannot be captured by GPU-only measurements alone.

# H ADDITIONAL RESULTS: NVML GENERALIZATION FOR ENERGY PREDICTION

Beyond examining NVML's limitations as a direct proxy for total energy, we also evaluated its generalization capabilities using leave-one-out cross-validation. Table 7 shows the results when NVML-based regression models are trained on all models of the same model family except one, then tested on the excluded model.

| Model | MAPE | Model | MAPE |
|-------|------|-------|------|
| Vicuna 7B | 48.3% | Llama 7B | 47.4% |
| Vicuna 13B | 51.1% | Llama 13B | 49.1% |
| Vicuna 33B | 50.0% | Llama 70B | 53.4% |
| Mistral 8B | 57.3% | Qwen 8B | 51.1% |
| Mistral 24B | 52.1% | Qwen 14B | 49.2% |
| Mistral 48B | 51.3% | Qwen 32B | 56.5% |

Table 7: Leave-one-out generalization results for NVML-based regression. Each row shows MAPE when using NVML measurements to predict total energy for a model excluded from training. The high error rates (47.4–57.3%) demonstrate NVML's poor generalization capabilities.

The results reveal significantly higher prediction errors compared to both in-sample NVML prediction (29.8-44.2% MAPE) and PIE-P's leave-one-out performance (15.8-24.5% MAPE). MAPE values range from 47.4% to 57.3%, with an average of 51.5% across all models (Table 7). These poor generalization results further reinforce our conclusion that GPU-only measurements are insufficient for accurate energy prediction in tensor-parallelized settings, particularly when generalizing to new model architectures or configurations. The substantial gap between PIE-P and NVML generalization performance (average improvement of 31.5 percentage points) demonstrates the critical importance of our approach's architecture-aware features and synchronization modeling.

# I ADDITIONAL ANALYSIS: GENERALIZATION ACROSS UNSEEN VARIANTS

Table 3 showed the leave-one-out generalization results across model variants. Here, we describe more analysis of the results beyond what was discussed in the main section.

**PIE-P generalizes better over batch sizes** PIE-P maintains consistent performance when trained on one batch size and tested on another. The MAPE values for batch size generalization range from 16.89% to 21.82%, with an average of 19.05% across all families. Interestingly, generalization from batch size 16 to 32 (19.33% average MAPE) performs similarly to generalization from batch size 32 to 16 (18.77% average MAPE), suggesting that PIE-P effectively captures the scaling relationship between batch size and energy consumption.

# J ABLATION STUDY : PIE-P WITHOUT ACCOUNTING FOR COMMUNICATION WAITING-PHASE IN TENSOR PARALLELISM

The purpose of this ablation is to isolate the contribution of *synchronization sampling*— our offline profiling of non-deterministic inter-GPU waiting during tensor-parallel collectives—to end-to-end energy prediction. In tensor model parallelism, small rank-to-rank skews around AllReduce introduce variable waiting that meaningfully contributes to communication energy; the question we answer here is how much accuracy is lost if this phenomenon is not profiled and modeled.

Methodologically, we keep the model-tree abstraction, structural (architecture) features, and aggregate runtime features identical to PIE-P, and we train/evaluate under the same data splits, hardware, and workloads; the *only* change is within the `AllReduce` module, where we disable the synchronization/wait-time component learned from offline sampling. Ground-truth energy is identical to the main experiments, and coarse meter misalignments are handled as in training for PIE-P; importantly, as in the main method, inference-time prediction incurs no measurement overhead. We report both model-level and module-level errors, with particular attention to the collective-communication nodes in the model tree.

| Excluded Family | PIE-P | PIE-P w/o waiting |
|-----------------|-------|-------------------|
| Vicuna | 24.1% | 41.4% |
| Mistral | 27.0% | 52.4% |
| Llama | 26.1% | 51.7% |
| Qwen | 27.6% | 55.0% |

Table 8: Cross-architecture generalization: MAPE when an entire family is excluded from training.

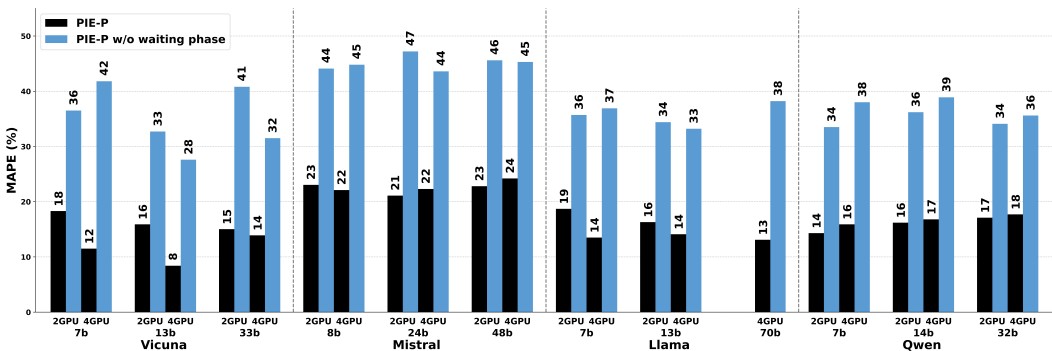

Figure 6: MAPE results for PIE-P across all models. The light blue bars show total energy prediction error without taking into account communication costs.

Empirically, eliminating the waiting-phase degrades accuracy substantially: the average model-level MAPE rises from **17.6%** for PIE-P to **36.9%** for the ablated variant—an absolute increase of **19.3** points ($\approx$**2.2** $\times$) (Figure 6). The degradation is most pronounced in settings where tensor-parallel communication dominates (e.g., larger models and 4-GPU runs), and is concentrated at collective nodes: AllReduce modules show the largest per-module errors, which then propagate upward in the model tree. Taken together, these results confirm that profiling synchronization variability once, offline, is necessary for robust, generalizable energy prediction in tensor-parallel inference and explains the consistent gap between the ablated baseline and PIE-P.

When holding out an entire architecture family, PIE-P maintains low error (24.1–27.6%) as seen in Table 8, whereas removing synchronization modeling (PIE-P w/o waiting) degrades accuracy by **17–27%**, underscoring that profiling rank-skew–induced waiting is essential for cross-architecture generalization.

## K ADDITIONAL RESULTS: FEATURE CORRELATION ANALYSIS

To assess the influence of different input features on total energy consumption, we performed a Spearman rank correlation analysis across all model configurations for Vicuna. As shown in Figure 7, GPU energy reported by NVML exhibits a strong correlation with total energy ($\rho$ between $0.633$ and $0.762$), as expected since GPU computation dominates energy consumption in LLMs.

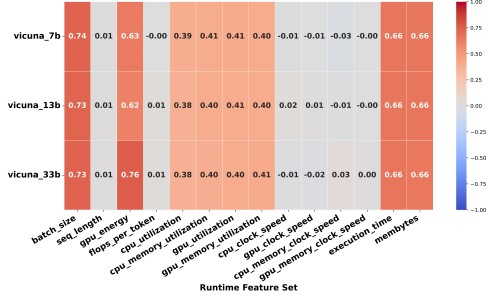

Figure 7: Spearman correlation heatmap of runtime features for the Total Model Energy of the 3 Vicuna models with the runtime features.

However, some other features also show substantial correlations with total energy consumption. Runtime features are particularly strongly correlated with total energy, with batch size ($\rho$ between $0.735$ and $0.737$), execution time ($\rho$ between $0.658$ and $0.664$), and Memory ($\rho$ between $0.656$ and $0.663$) all showing strong correlations.

These results demonstrate that relying solely on NVML-reported GPU energy is insufficient for accurate prediction, as system-level utilization metrics capture critical aspects of energy consumption—particularly in tensor-parallel configurations where communication and synchronization overheads are significant.

## L TRAINING METHODOLOGY

Our training methodology follows a hierarchical approach, first training module-level predictors and then combining them for model-level prediction. We employed 3-fold cross-validation at the module level, training on 70% of datapoints while testing on the remaining 30%.

Each module (e.g., Self-Attention, MLP, AllReduce) is sampled 10,000 times to build a robust dataset. A single sample corresponds to one aggregated measurement obtained by running the module 100,000 times in a specific GPU configuration setting. For each sampling run, we compute the mean, standard deviation, minimum, and maximum values of these features across all participating GPUs to account for runtime variance and workload imbalance. These are combined with structural features to form the complete feature vector for each sample.

To ensure that the models are evaluated under varied operational conditions, we perform sampling across multiple output configurations. Specifically, for each model variant, we run experiments using batch sizes of 8, 16, 32, and 64, and output sequence lengths of 512 and 1024. This sampling regime captures both low and high throughput scenarios, improving the robustness and generalizability of our predictors. We use vLLM (Kwon et al., 2023), a widely used inference serving python library to run our tests for all parallelism modalities.

For each model family (e.g., Vicuna), we combined module-level energy predictions across all variants using the extracted features. The model-level energy prediction was composed using the regression formula:

$$
\begin{aligned}
\text{model\_energy} = \mathcal{R}(&\text{self\_attention\_energy} \\
&+ \text{MLP\_energy} \\
&+ \text{AllReduce\_energy} + \ldots)
\end{aligned}
\tag{3}
$$

For the PIE-P without waiting phase baseline, we substituted the AllReduce module's predicted energy with just the network transfer predicted energy, excluding the waiting phase component. For the IrEne baseline, we excluded AllReduce energy completely from the regression.

Figure 2 reports the MAPE values obtained from this regression, where each model-level energy prediction MAPE is derived from a regression over its module-level energy components. The error bars represent the standard errors computed over the individual percentage errors of each prediction.

For each model variant, we obtain module-level energy MAPE values (in Table 5 by running each module type (e.g., Self-Attention, MLP, AllReduce) multiple times under varying input configurations and computing their predicted energy values. If a module appears multiple times in a model (e.g., multiple Feed Forward Networks), we first average the predicted energies across those instances. The final MAPE for a module type is then averaged across all variants and across all four model families.

Figure 8 shows the trade-off between actual energy consumption and inference time per token across Vicuna model sizes and GPU configurations.

To evaluate generalization across unseen model variants, we adopt a leave-one-variant-out strategy. Specifically, for each experiment, we exclude one model variant entirely from training and train a model-level energy regressor using the predicted module-level energies from the remaining variants within the same model family. During evaluation, we predict the total energy of the held-out variant using its module-level energy predictions as input. This setup allows us to assess the ability of PIE-P to generalize across model scales and configurations within a family.

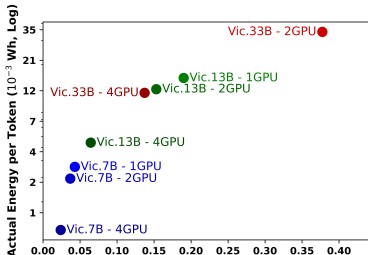

Figure 8: Ground truth energy trade-off between inference time per token and energy per token (on log scale) for Vicuna under tensor parallelism.

## M  GROUND TRUTH ENERGY VS INFERENCE TIME PLOT

Figure 8 shows the trade-off between actual energy consumption and inference time per token across Vicuna model sizes and GPU configurations. This is discussed in the context of generalization within the optimal model selection use case in Section 5.2.

# N   ABLATION: ROLE OF MODEL FEATURES

To assess the contribution of model structure features (e.g. attention heads, feed-forward dimension etc.), we perform an ablation by removing them from the prediction model and comparing MAPE with and without these features for all the variants of Vicuna. As shown in Table 9, while overall prediction accuracy remains similar in full-data settings, the inclusion of model features improves generalization performance in leave-one-out evaluations—reducing MAPE by 2–4% across Vicuna variants. These results suggest that model features help the regressors capture structural differences between variants, improving robustness when extrapolating to unseen configurations.

| Model Variant | With Model Features | Without Model Features |
|---|---|---|
| Vicuna 7B | 15.84% | 17.2% |
| Vicuna 13B | 17.72% | 18.2% |
| Vicuna 33B | 17.55% | 20.1% |

Table 9: Leave-one-out model-level prediction MAPE with and without model features.

Although the effect on raw prediction accuracy is modest, the improvement in generalization justifies their inclusion in the final model.