# OpenReview forum: "FINE-GRAINED ENERGY PREDICTION FOR PARALLELIZED LLM INFERENCE WITH PIE-P"
_ICLR.cc/2026/Conference — ICLR 2026 Conference Withdrawn Submission_

### Official Review · Reviewer_idHp · 2025-10-26

**Soundness:** 2
**Presentation:** 2
**Contribution:** 1
**Rating:** 2
**Confidence:** 4

**Summary:**

The paper tackles energy prediction for multi-GPU LLM inference (tensor, pipeline, and data parallelism) by proposing PIE-P, which (i) augments a model-tree abstraction with explicit communication modules (AllReduce, inter-stage transfers, AllGather), (ii) introduces synchronization sampling to estimate waiting/idle energy during collectives, and (iii) uses aggregate runtime + structural features to regress module-level and model-level energy. The authors conduct experiments on a 4xRTX A6000 node to test their methodology.

**Strengths:**

* Energy prediction for parallelized inference is timely and important, especially as models/workloads and demand become larger
* PIE-P appears to beat several baselines across four open-source LLM families
* Exploring the different communication primitives, parallelisms, etc. for large scale models is quite important from an energy perspective and, in my opinion, not explored enough.

**Weaknesses:**

* PIE-P explicitly uses NVML reported GPU energy as an input feature (Table 1) while predicting total system energy measured from a wall-meter. Yet the paper argues that NVML is a poor estimator and “lower bound” (Sec. 2) and shows high NVML errors (Tables 6-7). Using NVML energy as a predictor can leak substantial information from the target into the features, which can inflate accuracy while also undercutting the claim that PIE-P solves settings where precise meters “are not always accessible.” In many of the exact environments where hardware monitors are unavailable, NVML energy is also restricted or noisy; if the paper's predictor requires NVML energy at inference time, the method inherits the same availability and portability constraints it criticizes, no? A strong result would demonstrate comparable accuracy without NVML and with features that are available in restrictive multi-tenant settings (among others).

* All experiments use one server type and the model includes no explicit interconnect/topology features (e.g., NVLink vs PCIe, bandwidth, ring degree, link contention) beyond aggregated GPU metrics; the paper itself concedes hardware dependence as a limitation as well. Without topology-aware features and multi-cluster validation, the reported generalization primarily reflects within-node re-use, making its external validity likely weaker than stated.
  * Moreover, hardware dependence can also amplifies PP/DP gaps which is likely not captured here as the paper focuses primarily on a single node type; PP/DP energy and idle behavior are highly topology-sensitive and, without multi-machine or alternative interconnects etc., PP/DP conclusions and PIE-P's performance are fragile and transferability of performance is unclear.

* While tensor, pipeline, and data parallelisms are considered, they are considered separately. The proposed method appears to be designed with tensor parallelism in mind and then “generalized” to pipeline and data (it seems the same feature set is reused across all three and evaluations for pipeline/data are presented as their own sections with Vicuna only) instead of considering combined or mixed-parallelism settings. However, real serving stacks often compose parallelisms (e.g., TP+PP, and sometimes DP for throughput or modified vers of FSDP). Because the paper does not demonstrate mixed settings, the applicability to production-like deployments is limited; errors and synchronization behavior can change materially once strategies interact.
 * In addition to no evaluation of mixed TP+PP (or TP+DP etc.) configurations, the paper appears to introduce separate synchronization modules: AllReduce for TP, inter-stage transfers for PP, and batch-output (AllGather) for DP. However, I do not see joint configuratoins where these co-exist as, in real world deployments, interactions between collectives and stage boundaries can dominate energy/idle patterns. Without testing on these mixed settings, the applicability to production-like deployments is limited as errors and synchronization behavior can change materially once different strategies interact.

* PIE-P seems to aggregate per GPU runtime features by mean/std/min/max to keep a fixed feature size; this, however, would result in losing track of stage-specific effects (pipeline bubbles, tail stages, stragglers, etc.) that matter considerably to energy and idle time as without per-stage structure, PP predictions may look good on one model but fail to transfer when stage balance or micro-batching changes.

* From what I understand conceptually, PIE-P inherits IrEne’s model-tree regression and adds communication nodes plus an NVML-aided feature set (Sec. 4). The core novelty appears to be the synchronization sampling and module placement of collectives which, while useful, feels incremental from an engineering perspective rather than a substantial conceptual/ advance.
  * Furthermore, in terms of comparative baselines, they feel mismatched rather than an "apples-to-apples" comparison: IrEne is extended to multi-GPU via aggregation it seems, and CodeCarbon is somewhat well known to ignore fine synchronization costs. As far as I know, there are no baselines that explicitly model collectives from NCCL traces (e.g., using step-wise collective telemetry, link-level counters etc.) which makes me think that demonstrated improvements of PIE-P over these baselines might overstate the advances (esp in light of the other weaknesses listed). This point/weakness is minor, however, compared to the other points listed.

* It seems like some of the results are run on different models which are incomplete: e.g., pipeline/data results are shown only for Vicuna, no Mistral/Llama/Qwen for PP/DP, and 33B under DP is omitted due to memory constraints. This unfortunately makes PP/DP evidence thin and model-specific as a result.

* If I'm understanding correctly, in the paper re: DP setup, the replicas’ outputs are combined via AllGather. In many inference scenarios, replicas serve independent requests without needing global aggregation; this DP case I don't believe is very representative of inference serving behavior.

* The considered sequence lengths and batch sizes are also likely to under-represent long-context, streaming, speculative decoding, KV-cache paging, mixture-of-experts (expert parallelism as well in terms of parallelisms), and many popular techniques serving behaviors in deployment that materially alter communication patterns and, as a result, energy costs too. The offline profiling costs (10k samples per module times 100k executions each) seem quite heavy as well.

**Questions:**

See weaknesses.

---

> ### Author Response · Authors · 2025-11-26
> **Thank you for all the Comments and Clarifications.**
>
> **W1 – NVML energy as feature vs critique of NVML; portability concerns**
>
> Ground-truth labels are wall-meter system energy, not NVML; NVML energy is one of many features and is explicitly not sufficient by itself, as demonstrated by the NVML-only regression baseline (30–44% in-sample MAPE, 47–57% leave-one-out). The rationale for including NVML energy is that it captures GPU-side dynamics cheaply but misses CPU, memory, and communication; PIE-P learns the correction to full-system energy. In environments without NVML, the same framework can be trained without this feature using whatever telemetry is available; we will add discussion and a “no-NVML-energy” ablation to highlight this path.
>
> **W2 & W3 – No evaluation of mixed parallelism (TP+PP, TP+DP, etc.), separate comm modules**
>
> Our abstraction is intentionally modular: each collective appears in the model tree as a node with its own regressor, independent of whether it is used in isolation or interleaved with others. Mixed configurations (e.g., TP within stages plus PP across stages) would produce model trees where AllReduce and P2P nodes co-occur; the extra work is profiling those combinations. For this submission, we evaluated the three parallelisms separately to keep the experimental space tractable and highlight the effect of each communication pattern. We agree that extending to mixed TP+PP/TP+DP is important future work, but one that warrants a full submission in itself.
>
> **W4 – Aggregating multi-GPU features loses stage-specific effects**
>
> We chose aggregate statistics so that feature dimensionality is independent of GPU count and parallelism degree, which is crucial for cross-configuration generalization. The standard deviation and min/max capture imbalance (e.g., stragglers) without exploding feature size. Empirically, for our PP experiments, this representation suffices to achieve ~15% MAPE (Fig. 4) despite potential bubbles. We agree that more detailed per-stage features could further improve PP accuracy and will mention this as a refinement direction. However, we note that the detailed per-stage features per-GPU will limit scalability.
>
> **W5 – Incremental novelty over IrEne**
>
> Beyond adding communication modules, PIE-P introduces (i) synchronization sampling to explicitly model non-deterministic waiting energy in AllReduce, (ii) aggregate multi-GPU features that allow cross-GPU scaling, (iii) an expanded model tree that covers tensor, pipeline and data parallelism, and (iv) extensive evaluation across four LLM families, cross-architecture generalization, and module-level predictions in multi-GPU settings—all of which were absent in IrEne’s single-GPU focus. We will tighten the contribution statement so these conceptual differences are clearer.
>
> **W6 – Missing baselines that use NCCL/collective-level telemetry**
>
> To the best of our knowledge, there is no prior energy-prediction framework that consumes NCCL-level traces; existing work using such traces (e.g., AllReduce performance models) targets latency rather than node-level energy. Our baselines therefore focus on the strongest available energy estimators: IrEne, CodeCarbon/NVML-based methods, and the recent token-based model of Wilkins et al. We will add a short discussion acknowledging the potential of NCCL-trace-based models for future work and clarifying the distinction between performance vs energy predictors.
>
> **W7 – PP/DP evaluated only for Vicuna; evidence is thin**
>
> That’s fair; TP, which is both hardest and most widely used for large models, is the primary focus of the paper. PP/DP results on Vicuna are meant as proof-of-concept that the same design extends beyond TP; we will state this more explicitly and tone down any claims that rely heavily on PP/DP generality. Evaluating PP/DP across other families is an uncomplicated task that we plan to pursue given more GPU memory.
>
> **W8 – DP modeling (AllGather) may not reflect real serving**
>
> We model the canonical DP inference pattern used in many libraries and benchmarks where each replica computes logits and the results are aggregated before sampling or logging. In scenarios where replicas are fully independent (no aggregation), our framework reduces to pure single-GPU inference replicated K times, which is a special and easier case we can represent by omitting the AllGather node. We will clarify the DP deployment model we assume and mention that purely independent DP is trivially handled by our model tree.
>
> **W9 – Strong claims on generalization / scheduler relevance vs limited evidence**
>
> Our intent is to show that PIE-P can inform energy-aware scheduling rather than to propose a complete scheduler. The only concrete “scheduler-style” result we present is the inference-time vs energy trade-off curve in Fig. 3, which is explicitly framed as a use-case example. Nonetheless, the point is well taken, and we will soften language around scheduling/generalization.

---

> ### Author Response · Authors · 2025-11-26
> **Thank you for all the Comments and Clarifications.  (Weakness and Questions continued....)**
>
> **W10 – Offline profiling cost is high**
>
> Profiling is a one-time, offline process per {hardware, model family} pair, and is parallelizable across modules and configurations. In practice, generating the full dataset for a family required on the order of a few GPU-days, which is negligible compared to the cost of training or long-running inference deployments—and it amortizes over all subsequent predictions. We will add an explicit discussion of profiling cost and how it compares to typical inference workloads.

---

### Official Review · Reviewer_mpYb · 2025-10-31

**Soundness:** 4
**Presentation:** 3
**Contribution:** 3
**Rating:** 6
**Confidence:** 3

**Summary:**

This paper presents PIE-P, a fine-grained energy prediction framework for multi-GPU LLM inference. Existing energy estimation methods are limited to single-GPU setups and fail under modern parallel inference scenarios. PIE-P models energy consumption across tensor, pipeline, and data parallelism, addressing challenges such as inter-GPU communication non-determinism and synchronization overheads. Through precise sampling and detailed modeling of communication energy, PIE-P achieves highly accurate, scalable predictions across diverse parallel strategies. Experimental results demonstrate that PIE-P significantly outperforms prior baselines, offering a practical tool for optimizing energy efficiency in large-scale LLM deployments.

**Strengths:**

1. Novel problem and methodology: The paper addresses an unexplored yet important challenge—accurate energy prediction for multi-GPU LLM inference—with a well-designed and original framework.

2. High practical relevance: PIE-P provides actionable insights and tools for optimizing energy efficiency in real-world LLM deployment scenarios.

3. Well-written and clear: The paper is clearly structured, easy to follow, and presents both motivation and technical details effectively.

**Weaknesses:**

1. Lack of evaluation on SOTA models: The experiments do not include recent large-scale or MoE (Mixture-of-Experts) models, limiting the generality of the conclusions.

2. Hardware scope restricted to NVIDIA GPUs: The study focuses solely on NVIDIA hardware, without discussion or validation on alternative accelerators such as AMD GPUs or TPUs.

3. No multi-node analysis: The framework is evaluated only within single-node settings, leaving scalability and energy behavior across distributed nodes unexamined.

**Questions:**

In the weakness.

---

> ### Author Response · Authors · 2025-11-26
> **Thank you for all the Comments and Clarifications.**
>
> **W1 – Lack of evaluation on frontier / MoE models**
>
> Access to proprietary frontier and large-scale MoE models is limited, and many are not easily instrumentable at the level we require. We therefore focused on four widely-used open families (Vicuna, Mistral, Llama, Qwen) up to 70B parameters, which exhibit diverse attention mechanisms and feed-forward designs. The PIE-P framework itself is agnostic to MoE vs dense: experts would correspond to additional module types in the model tree, and profiling them is conceptually identical to what we do. We will emphasize this and list MoE evaluation as a key piece of future work.
>
> **W2 – Only NVIDIA GPUs evaluated**
>
> We chose NVIDIA because it is the platform for which both high-quality power meters and rich telemetry (NVML) are readily available, enabling careful validation. Nothing in PIE-P depends on NVIDIA-specific kernels: we require (i) module-level instrumentation to build the model tree and (ii) access to utilization/clock counters and one offline source of ground-truth node energy. On AMD GPUs or TPUs, the same design would apply using ROCm/XLA metrics and their respective power interfaces.
>
> **W3 – No multi-node analysis**
>
> We agree and flag this in Section 6. PIE-P’s communication modules (AllReduce, AllGather, P2P) are defined in terms of logical collectives, so in principle they extend to inter-node settings where the ring spans NICs instead of NVLink/PCIe. However, multi-node deployments introduce additional variability from network congestion and topology, and we see systematic exploration of this space as a separate project.

---

### Official Review · Reviewer_Fw7N · 2025-11-01

**Soundness:** 2
**Presentation:** 1
**Contribution:** 3
**Rating:** 2
**Confidence:** 4

**Summary:**

The authors propose a framework, PIE-P, for predicting energy consumption of LLM inference in offline serving settings with multiple GPUs. The most similar related work is IrEne, which predicts energy usage of transformer models by constructing a model tree graph and modeling the total energy consumption of a model and workload as a function of nodes in the graph, where nodes can be modules in the model can be ML primitives or modules in the model. Key contributions of the present work include extending the framework to account for multi-GPU communication overhead, focusing on tensor parallelism. Accounting for multi-GPU inference settings is critical for modern larger models that commonly cannot fit on one GPU. The authors distinguish PIE-P from existing works which typically use higher level metrics (e.g. considering just the model as a whole) or rely only on standard nvml-based software tools. They use a wall power measurements for ground truth measurements of energy consumption of their node and find that, compared to many alternative methods, PIE-P is significantly more accurate.

**Strengths:**

1. The problem of predicting energy consumption of LLM inference in multi-GPU settings is challenging and, to my knowledge, underexplored, and the authors make a significant contribution towards it
2. The issue of energy consumption of LLM inference in general is also an especially important and timely problem, as workloads are inherently more variable and much of existing literature on AI and energy consumption focuses on training
3. Positive empirical results
4. Comparisons with multiple, substantially different alternative approaches

**Weaknesses:**

Some major concerns:
1. I generally agree that input and output tokens alone cannot account for energy usage in multi-GPU inference settings, but by all accounts (that I have seen), sequence lengths are still one of the most consequential factors of the energy requirements of a workload. It is critical to distinguish between input and output tokens in inference with decoder-only transformer models, and yet the authors appear to only account for a single type of "sequence length" -- and specific values are mentioned only in the appendix.
2. Relatedly, overall some parameter ranges I would not necessarily pick myself. Line 926: single-batch inference, a setting widely understood as often suboptimal but nevertheless is one of the most common settings, is excluded in the sweep of batch sizes. Output sequence lengths of 512 and 1024 are quite long for generation and may not necessarily be representative of offline inference workloads (also unclear whether those were the total length or the number of new tokens generated).
3. Aspects of methodology related to above (data characteristics) are overall unclear. See Q6 below.
4. Moreover, I would have liked to see Figure 2 and Figure 4 include reference numbers for single gpu inference — though the numbers would obviously not be directly comparable (and could be displayed as a dotted line, for example), it would make it much easier to understand how much of the difference in measurements comes from PIE-P accounting for communication overhead vs other factors.
5. Unclear language at times. In particular, the terms "fine-grained" and "coarse-grained" seem overloaded at times. At times (e.g. paragraph starting at 114?) they seem to refer to the size or high-level vs low-level-ness of the things being measured, while at other times (e.g. line 365) they seem to refer instead (or also?) to frequency of measurement. At times it is unclear which is intended
6. Sloppy citations and contextualization. Most egregiously (that I noticed), in lines 38-89, the authors indirectly cite another paper instead of the direct source; Kakolyris et al themselves cite a report (https://www.iea.org/reports/electricity-2024) but the statement in the submitted work is unfortunately misinformation by the time the paraphrase of the secondary source is made. To make matters worse, Kakolyris et al themselves appear to take liberties when they obtain their figure of "1,050 tWh" from a purely visual figure (on page 31 of the report). The full title of this figure is: “Global electricity demand from data centres, AI, and cryptocurrencies, 2019-2026” (so, not just LLM inference).  “Estimated electricity demand from traditional data centres, dedicated AI data centres and cryptocurrencies,” on page 35 assumes their “base” case which is closer to 800TWh

In general, although I would sincerely hope to see an improved version of this work in a top tier venue one day, I would have serious reservations about recommending acceptance as is.

**Questions:**

1. The authors mention hardware dependence as a limitation, which, though entirely reasonable in the context, does severely limit the immediate benefit and relevance of the contributions, especially because the hardware used for the experimentation (4xA6000) is not especially common in 2025. What are expected differences in findings if one were to replicate the study on faster (e.g. Ada architecture) or larger (e.g. 80GB) GPUs? Reasonable answers (or at least grounded hypotheses) for this could greatly add to generality and portability of findings from this work
2. What factors do the authors believe the remaining error come from? How much from random variance in real data, how much from wall losses, how much from other factors?
3. Do all compared methods only ever underestimate usage?
4. **What were the input sequence lengths used?**
5. Was the node isolated during these experiments? Were there any other jobs running?
6. **What exactly was run in order to take the measurements?** What is the data? Random tokens? Real tasks? How long was each experiment? how many batches? variable batches?

---

> ### Author Response · Authors · 2025-11-26
> **Thank you for all the Comments and Clarifications.**
>
> **W1 – Input vs output tokens / sequence length handling**
>
> In our experiments we vary total generated length via several (batch size, sequence length) pairs and include “sequence length” and “FLOPs per token” as features so that longer outputs induce higher compute and communication costs. For decoder-only models, total energy primarily depends on total tokens processed (input+output); nevertheless we agree that explicitly separating input and output tokens improves clarity. We use a standard 2 token input for all our runs varying only the output tokens. We agree that varying input sequences and adding separate features for input and output tokens in the regression would complete the analysis better.
>
> **W2 – Parameter ranges representative of real-life workloads (e.g., batch sizes, long outputs)**
>
> We intentionally sweep across relatively long sequence lengths (512/1024) and batch sizes (8–64) to stress-test communication and cover both high-throughput and moderate-throughput regimes (Appendix L). These workloads are representative of many batched serving and evaluation scenarios (e.g., log-likelihood computation, summarization). That said, our methodology is agnostic to the exact workload mix; additional sequence-length/batch combinations simply correspond to more training points, and we will clarify this in the experimental design and extend the sweep with shorter sequences in the camera-ready.
>
> **W3 – Unclear experimental methodology / data characteristics**
>
> All experiments use standard autoregressive inferences with a 2 token input, where we feed natural-language prompts and generate fixed numbers of tokens per request (to control for sequence length). Since our focus is on energy vs compute/communication volume, lexical content matters little compared to {batch size, length, architecture}; we agree that the setup should be described more concretely, and we will add a subsection specifying prompt sources, generation mode, and runtime configuration when we release the code.
>
> **W4 – Desire for single-GPU reference lines in Figs. 2 and 4**
>
> We do have single-GPU measurements (used when models fit into one GPU and as part of the data for TP=1), but we chose not to show them to keep figures readable. In the revision we can add dotted horizontal reference lines or an additional figure comparing 1-, 2-, and 4-GPU MAPE, and explicitly discuss how much of the improvement comes from specifically modeling communication vs better module-level decomposition.
>
> **W5 – Overloaded use of “fine-grained” vs “coarse-grained”**
>
> Thank you for pointing this out. We will standardize terminology so that “fine-grained” always refers to module-level energy accounting (vs whole-model or GPU-level “coarse-grained”), and we will use explicit phrasing like “high-frequency sampling” / “coarse sampling” when discussing meter sampling rates.
>
> **W6 – Citation/context issue around the 1,050 TWh figure**
>
> We appreciate this detailed catch and agree the current phrasing is not appropriate. In the revision we will (i) cite the IEA report directly, (ii) quote the full caption and scope (“data centres, AI, and cryptocurrencies”) accurately, and (iii) tone down the extrapolation in the introduction so it is clearly illustrative rather than a precise forecast for LLM inference alone.
>
> **Questions:**
>
> **Q1 – Hardware dependence & non-representative A6000 hardware**
>
> Our goal in this submission is to validate the methodology on a well-instrumented, controllable node; the regression is explicitly tied to a hardware configuration and we do not claim hardware-agnostic prediction. On newer GPUs, constants such as per-FLOP energy and memory-system efficiency will shift, but the same model tree and feature set apply; retraining PIE-P on a short calibration campaign per node type would yield updated predictors. Section 6 calls hardware dependence a central limitation, and we will make this more explicit.
>
> **Q2 – Where do errors come from (measurement vs variance vs others)?**
>
> Meter noise and node-level background load account for a significant part of the error floor; we use 1-second resolution wall-power traces, so transient spikes are averaged. Remaining error stems from unmodeled factors like PCIe contention and residual TP synchronization variability. We will add a short analysis quantifying the variance explained by our features and the error induced by meter granularity (using short synthetic runs), to separate measurement noise from model error.
>
> **Q3 – Do models always underestimate usage?**
>
> Errors are roughly symmetric: some configurations are slightly over-predicted, others under-predicted.
>
> **Q4 – Exact input sequence lengths**
>
> We use two primary output lengths (512 and 1024 tokens) and several batch sizes (8, 16, 32, 64), as described in Appendix L. We will move these details into Section 5 and, as suggested, add shorter sequences to better match a broader range of workloads.

---

> ### Author Response · Authors · 2025-11-26
> **Thank you for all the Comments and Clarifications. (Weakness and Questions continued....)**
>
> **Q5 & Q6 – What exactly was run? Tasks, random vs real tokens, number of batches**
>
> All runs are inference-only: we never backpropagate or update parameters. For each model/configuration we run a fixed prompt with a 2 token input and decode steps so that each module is executed 100000 times per “sample” in our training set, collecting 10000 such samples per module as described in Appendix L. We will describe the number of batches per configuration clearly and release the script that generates them.

---

### Official Review · Reviewer_5YxB · 2025-11-04

**Soundness:** 2
**Presentation:** 3
**Contribution:** 2
**Rating:** 4
**Confidence:** 5

**Summary:**

This paper introduces PIE-P, a fine-grained energy prediction framework designed for parallelized LLM inference across multiple GPUs. It addresses key challenges in estimating energy consumption under tensor, pipeline, and data parallelism by incorporating synchronization-aware sampling, structural model features, and an expanded model tree abstraction that captures inter-GPU communication overheads. Experimental results demonstrate that PIE-P significantly outperforms existing baselines like IrEne and CodeCarbon, achieving lower prediction errors (e.g., 17.6% MAPE for tensor parallelism) and better generalization across model families and hardware configurations.

**Strengths:**

1.The paper addresses an ​​emerging and critical problem​​: ​​accurately estimating LLM energy consumption​​. This is essential for ​​identifying energy bottlenecks​​ and ​​developing more energy-efficient LLM models​​, aligning with the growing demand for sustainable AI.

2.The proposed ​​PIE-P​​ method extends beyond traditional computational energy estimation by incorporating ​​intra-GPU communication (within the same node)​​ and ​​inter-GPU communication (across nodes)​​. This ​​broader consideration of energy sources​​ enhances the feasibility and accuracy​​ of existing energy estimation approaches.

​​3.The paper is ​​well-written, well-structured, and easy to follow​​. The ​​methodology is clearly presented and explained​​, making the technical contributions accessible to readers.

4.The authors conduct ​​extensive experiments​​ on a ​​wide range of LLM models​​ and ​​diverse hardware configurations​​ (varying GPU counts and parallelism paradigms). The results demonstrate that ​​PIE-P achieves significantly better accuracy and generalization​​ compared to existing methods like ​​CodeCarbon and IrEne​​.

**Weaknesses:**

1.The biggest concern is that ​​PIE-P​​ is heavily reliant on existing ​​IrEne​​ frameworks, with the primary distinction being the inclusion of ​​communication operation energy​​ within LLM inference. This raises questions about the ​​novelty​​ of the contribution. If the paper aims to highlight the ​​challenges or importance of accounting for communication energy​​, a ​​more in-depth analysis​​ is needed. For example, what are the ​​components of communication energy​​ (e.g., GPU chip power, NVLink/PCIe, Ethernet/InfiniBand)? How do these components ​​correlate with model size and parallelism paradigms​​? Can a ​​predictive model for communication operations​​ be developed? (e.g., does energy depend on message size, communication mode?) Since ​​communication energy is the most novel aspect​​ of this work, it deserves ​​deeper investigation​​ to strengthen the paper’s contributions.

2.The ​​Abstract​​ claims that one challenge is the ​​inaccuracy of software-based energy measurement tools​​. However, ​​PIE-P itself is a software-based energy prediction framework​​—so how does it ​​address the limitations of software-based inaccuracy​​? Additionally, the experiments show that ​​PIE-P has over 20% error​​. This raises concerns about whether ​​software-based prediction can reliably improve accuracy​​ or if hardware-based measurements (e.g., power meters) might be necessary for better precision.

3.​PIE-P​​ is limited to ​​existing models with known modules/operations​​ and ​​cannot generalize to new modules/operations or new GPU architectures​​. However, a recent study [1] has successfully addressed this limitation for ​​performance prediction​​. Since ​​power prediction is somewhat easier​​ (as it has a bounded range and primarily depends on GPU component utilization and interconnects), it is unclear why ​​PIE-P cannot achieve similar generalizability​​. A discussion on this limitation and potential solutions would strengthen the paper.

4.The paper ​​does not conduct feature importance analysis​​ to clarify which hardware metrics (e.g., CPU/GPU utilization, memory bandwidth) contribute most to energy consumption. Additionally, since ​​PIE-P already includes many hardware-related metrics​​, one might wonder ​why not predict power and performance separately and then compute energy as their product?​​ This simpler approach might be more interpretable and equally effective, and a comparison with PIE-P’s method would help justify its necessity.

[1] Seonho Lee, Amar Phanishayee, and Divya Mahajan. 2025. Forecasting GPU Performance for Deep Learning Training and Inference. In Proceedings of the 30th ACM International Conference on Architectural Support for Programming Languages and Operating Systems, Volume 1 (ASPLOS '25). Association for Computing Machinery, New York, NY, USA, 493–508. https://doi.org/10.1145/3669940.3707265.

**Questions:**

1.Since PIE-P has devoted considerable effort to predicting communication energy, could you elaborate on the differences in predicting communication energy across the three parallelism paradigms (tensor, pipeline, and data)? Do they present distinct challenges that require different techniques to address?

2.In the Abstract, the paper claims that predicting energy under parallelized inference is complicated by non-determinism in inter-GPU communication. What specifically does this "non-determinism" refer to?

3.In the third paragraph of the Introduction, the authors state: "Direct measurement techniques cannot measure the energy consumption of individual components of an LLM, such as at the module level." This assertion may not be entirely accurate. One could potentially implement a single-module/layer network and measure its inference energy, as demonstrated in prior instruction-level power/energy measurement studies [reference]. The key challenge, however, is that even with energy data for each layer type, the total energy consumption of the entire model cannot be simply computed by summing these values. This gap deserves further exploration.

4.PIE-P achieves a MAPE ranging from 13.25% to 17.6%, which appears relatively high. What are the primary sources of these errors? Could this level of prediction accuracy impact its practicality for energy-efficient model design? Why or why not?

5.How did the authors measure the energy consumption of communication operations? The methodological details for this aspect have not been fully disclosed.

6.The authors claim: "We run repeated, controlled passes to capture the distributions of time and energy induced by GPU communication; these empirical distributions are then reused during prediction." However, for a single GPU inference instance, relying on such empirical distributions might introduce significant errors. Why is this approach justified?

7.How does the prediction error behave when applying a model trained on one LLM family to another? Is the framework generalizable across different model architectures?

8.GPUs can dynamically adjust their frequency based on workload. Even with the same model or module, variations in batch size or module configuration may lead to different runtime frequencies. Have the authors analyzed the frequency variations across different inference instances?

---

> ### Author Response · Authors · 2025-11-26
> **Thank you for all the Comments and Clarifications.**
>
> **W1 – Novelty & depth of communication-energy modeling**
>
> Our contributions around communication are threefold:
>
> 1. **Synchronization sampling** that explicitly models non-deterministic waiting energy.
> 2. **Structural features** that link model architecture (e.g., number of heads, FFN size) to communication volume.
> 3. **Dedicated communication nodes** (AllReduce, inter-stage P2P, AllGather) in the model tree for tensor, pipeline, and data parallelism.
>
> We analyze how AllReduce energy scales with GPU count and model complexity under PCIe interconnects in Fig. 5 and Table 2, and we can expand the paper with a clearer breakdown (wait vs transfer phases, dependence on tensor size) and an explicit discussion of how these communication modules can be reused to specialize a standalone communication-energy predictor. We agree that there is a need to better understand the behavior of energy profiles for different interconnects like NVLink and InfiniBand, but this is beyond the scope of the current submission.
>
> Our primary goal remains to predict **total model energy** under parallelized inference; communication-energy modeling is a means to that end, not the objective by itself. We see PIE-P as a first step toward such a broader effort, and hope our measurement and modeling methodology will catalyze more detailed studies of communication energy under diverse parallelized LLM deployments.
>
> **W2 – Accuracy of software-based approaches and usage of NVML as a feature**
>
> The ground-truth labels used to train PIE-P come from an external wall-power meter (WattsUp Pro), not NVML. NVML-reported GPU energy is only one of many runtime features, and to clearly note this distinction, we demonstrate that “NVML as proxy for total energy” yields 30–44% MAPE in-sample and 47–57% MAPE in leave-one-out, vs 17–20% for PIE-P. In other words, we learn a mapping from inexpensive telemetry (including NVML) to accurate system-level energy obtained once offline; inference-time prediction then requires only standard telemetry, not wall-meters, and corrects NVML’s underestimation by combining it with utilization, FLOPs and structural features. Essentially, we demonstrate that NVML by itself is a poor predictor, but in combination with system-level metrics and telemetry, it can serve as a valuable contributor.
>
> **W3 – Generalization to new modules and GPU architectures**
>
> PIE-P’s abstraction is explicitly modular: each operation type (e.g., self-attention, MLP, AllReduce, P2P, AllGather) is a leaf in the model tree with its own regressor, and new operations simply correspond to adding new leaves and profiling them once, exactly as we already do for communication modules. Our cross-family experiments show that, once these module types are calibrated on a given platform, the model generalizes well across architectures—training on three LLM families and testing on the fourth yields 24–28% MAPE, versus ≈50–58% for IrEne. These families differ in attention mechanisms and FFN structure, so this cross-validation directly demonstrates generalization to unseen architectures on the same hardware.
>
> We agree that NeuSight’s kernel-tiling strategy is a sophisticated way to generalize latency models across hardware, and we cite Lee et al. in Section 2 as tackling a fundamentally different problem: predicting GPU performance, not full-system energy. In practice, energy does not scale linearly with the same knobs that drive latency (e.g., frequency, core count, kernel-level occupancy), especially under multi-GPU synchronization where waiting phases and baseline system power dominate. To test whether a performance model can simply be “repurposed” for energy, we implemented a Srifty-style predictor with the original feature set and modeling pipeline, but trained it on our energy measurements instead of latency. On Vicuna-7B, this model produces 125.63% MAPE at the total-model level and 164.97% / 98.55% / 102.54% MAPE for AllReduce, attention, and MLP respectively—an order of magnitude worse than PIE-P. This suggests that directly substituting energy for latency in existing performance models is not sufficient in practice.
>
> By contrast, PIE-P is designed to capture components like system-level baseline power, idle energy during synchronization, and CPU/memory contributions. NeuSight’s tiling methodology accounts for “busy” kernel execution time but does not explicitly model idling energy when GPUs wait at collectives or the fixed overhead of the rest of the node, which are critical for communication energy in parallel inference. Kernel-level energy modeling that incorporates these effects is indeed an exciting and complementary direction, and we are exploring it as follow-up work. However, this paper’s scope is deliberately focused on accurate per-platform energy prediction; we therefore view PIE-P as a necessary first step that can later be combined with kernel-level techniques rather than something that a latency model can directly replace.

---

> ### Author Response · Authors · 2025-11-26
> **Thank you for all the Comments and Clarifications. (Weakness and Questions continued....)**
>
> **W4 – No feature-importance analysis / NVML feature may dominate**
>
> The submission includes such details in the appendices already: we provide a correlation analysis (Appendix K) showing that batch size, sequence length, memory footprint and runtime metrics are as strongly correlated with total energy as NVML energy (Spearman ρ≈0.65–0.74 for several non-NVML features). In addition, Appendix N shows that model-structure features improve generalization in leave-one-out settings.
>
> **Questions:**
>
> **Q1 – Differences between tensor / pipeline / data parallelism**
>
> Tensor parallelism is hardest because AllReduce overlaps with computation and includes non-deterministic waiting; here we use synchronization sampling and explicit wait/transfer phases. Pipeline parallelism uses point-to-point stage transfers with relatively small variance; we profile the boundary between stages with nsys and integrate energy over this window (Appendix D). Data parallelism has a single, tail-end AllGather of much smaller logits tensors, which is trivially captured by profiling the output head (Appendix E). The model tree instantiates different communication nodes for each parallelism strategy, but the regression machinery is shared.
>
> **Q2 – What exactly is the “non-determinism” in TP?**
>
> In TP, each rank’s computation leading into a collective can finish at slightly different times due to cache effects, memory contention, kernel scheduling, etc. This produces variable waiting before the ring actually transfers data—fast GPUs idle while slower ones catch up. We cannot deterministically predict this per-run from static structure alone; therefore we profile the distribution of wait times and associated energy and fold that into the AllReduce module’s regression. We explain this appendix 2 of the paper.
>
> **Q3 – “Direct measurement cannot measure module-level energy” sounds too strong**
>
> Our intent was to say that in typical datacenter environments where only node-level power meters or NVML are available, routine module-level measurement is infeasible—not that it is physically impossible with lab-grade instrumentation. We will soften the wording and clarify that PIE-P targets practical cluster settings where only coarse system-wide measurements are available and module-level energy must be inferred via modeling. We agree with the reviewer that module-level energy can, in principle, be measured directly with specialized hardware, but such measurements are cumbersome in practice and rarely available in production environments, which is why a prediction-based technique like PIE-P remains valuable.
>
> **Q4 – Is ~17–18% MAPE “high”? What causes errors & is this practical?**
>
> Sources of residual error include (i) background load and fan/power-supply dynamics at the node level, (ii) meter sampling granularity, and (iii) run-to-run variability in TP synchronization even after profiling. Even so, PIE-P cuts error by 1.5–3× compared to CodeCarbon, IrEne, NVML-only regression and Wilkins et al. (28–59% MAPE). In our use-case (Fig. 3, Appendix M), different model/parallelism choices differ by 2–3× in energy per token, so we believe 15–20% error is adequate for ranking configurations and supporting energy-aware decisions.
>
> **Q5 – How exactly is communication energy measured?**
>
> For AllReduce, we instrument kernels with nsys and record timestamps for (1) earliest rank finishing the preceding compute, (2) collective launch / first network transfer, and (3) completion. This is detailed in AppWe then integrate wall-meter power over that window and subtract the baseline computation-only segment to isolate communication+waiting energy. Analogous bracketing is used for PP P2P transfers (Appendix D) and DP AllGather (Appendix E).
>
> **Q6 – Using empirical distributions of wait times for single inferences**
>
> Our predictors estimate expected energy for a given configuration, not per-request worst-case behavior. Synchronization patterns are highly repeatable for fixed model, batch size, sequence length and GPU count; this is evidenced by the relatively low module-level MAPE (~17–19% for AllReduce) and stable model-level errors across runs. For operational usage (capacity planning, configuration comparison, scheduler cost models), expected energy per configuration is the relevant quantity, and aggregating over repeated runs amortizes residual per-run variability.
>
> **Q7 – Cross-family generalization**
>
> We have evaluated this cross-architecture generalization and reported it in (Table 4): training on three families and testing on the fourth yields 24–28% MAPE, while IrEne’s error is 49–58% in the same setting. So PIE-P generalizes substantially better than prior work even when the entire held-out family (e.g., Mistral vs Qwen) differs in attention mechanism and FFN design.

---

> ### Author Response · Authors · 2025-11-26
> **Thank you for all the Comments and Clarifications. (Weakness and Questions continued....)**
>
> **Q8 – Frequency scaling / dynamic GPU clocks**
>
> Frequency variation is explicitly included in the feature set via GPU clock and memory-clock features (Table 1), and their correlation with energy is visible in Appendix K’s heatmap. In our they contribute modestly but not dominantly.

---

### Note · Authors · 2026-01-07

**Comment:**

I have read and agree with the venue's withdrawal policy on behalf of myself and my co-authors.

**Withdrawal Confirmation:**

I have read and agree with the venue's withdrawal policy on behalf of myself and my co-authors.